# Preparation and Mechanism of Bio-Based Sodium Alginate Fibers with Flame Retardant and Antibacterial Properties

**DOI:** 10.3390/polym15010154

**Published:** 2022-12-29

**Authors:** Jiamin Xu, Zhenlin Jiang, Fang Hou, Keyu Zhu, Chenxue Xu, Chaosheng Wang, Huaping Wang

**Affiliations:** 1College of Chemistry and Chemical Engineering, Research Center for Advanced Mirco- and Nano-Fabrication Materials, Shanghai University of Engineering Science, Shanghai 201620, China; 2Science and Technology on Advanced Ceramic Fibers and Composites Laboratory, National University of Defense Technology, Changsha 410073, China; 3Key Laboratory of High Performance Fibers & Products, Ministry of Education, Donghua University, Shanghai 201620, China

**Keywords:** sodium alginate fiber, phytic acid, DL-arginine, flame retardant, antibacterial

## Abstract

Flame retardant and antibacterial sodium alginate (SA) fiber were fabricated using the bio-based flame retardant of phytic acid and DL-arginine successively, and then the morphological structures, combustion behavior, thermal stability, and mechanical as well as antibacterial properties of SA fiber were investigated carefully. It is found that when the additional amount of PADL (reaction products of phytic acid and DL-arginine) in SA composite fiber is 20 wt%, its limiting oxygen index (LOI) is 40.0 ± 0.3%, and UL−94 is V−0 grade. The combustion behavior of composite fiber shows that PADL can effectively reduce combustion heat and promote carbon formation. Its peak of HRR (pkHRR) is 5.9% of pure SA fiber, and the residual carbon increases from 23.0 ± 0.1% to 44.2 ± 0.2%. At the same time, the density of the residual carbon increases gradually. PADL can promote SA to form expanded carbon with increasing density, and isolate the heat and volatilization of combustible gases. The guanidine group of DL-arginine can interact with the cell membrane to kill bacteria, and the antibacterial property of SA composite fiber is increased by 30%. This study provides a very ecological, safe, environmentally friendly and simple method to prepare flame retardant and antibacterial SA composite fiber with bio-based materials.

## 1. Introduction

Alginate is a natural linear polysaccharide polymer extracted from seaweed. It has excellent biocompatibility, low toxicity, good moisture absorption and degradability [1,2,3,4,5]. It is widely used in textiles, food and medicine, and is an ideal natural biological resource for human beings [6,7,8,9,10]. The alginate prepared by wet spinning is a pure natural textile [11,12,13,14,15], which has excellent skin-friendly, moisture absorption and environmental protection characteristics, and is widely used in medical and household textiles and other materials. However, the polysaccharide structure of alginate materials inevitably has problems of a limited flame retardancy and low antibacterial grade. The application of alginate materials in the textile field still needs to improve its flame retardancy, and, in particular, their flame retardancy and antibacterial properties have been the focus of researchers [16,17,18,19,20,21,22,23,24,25,26,27,28].

Physical and chemical modifications are commonly used methods for sodium alginate at present. Standard methods include wet spinning and electrospinning. Electrospinning forms polymer solution into nanofibers through the effect of the external electric field, which is commonly used in the medical field to prepare wound dressings [29,30,31]. Wet spinning is more flexible and suitable for producing small batches of high-value-added differential fibers than electrostatic spinning. The physically modified composite sodium alginate materials, which are usually prepared by mixing and dispersing functional modifiers, are greatly affected by the dispersion of modifiers in the composite materials, resulting in a large number of additives and an enormous impact on the mechanical properties of the materials. Wang et al. used polyvinyl alcohol (PVA) as the modifier to prepare composite fibers by wet spinning. Using the hydrogen bond structure between PVA and sodium alginate, the mechanical properties of the fibers were improved while ensuring a certain flame retardancy of the composite fibers [32]. Li et al. prepared a highly flame retardant blend fiber by blending the flame retardant alginate fiber with polyester fiber. The flammability of the blended fiber meets the requirements for filler in the combustion chamber and can be used as the filler for quilts, toys, sleeping bags and pillows [33]. The typical chemical modification methods are oxidation modification and esterification modification. Ding et al. prepared a series of periodate oxidized sodium alginate (OSA) by oxidative modification using natural polysaccharides as crosslink agents. They found that the molecular weight of OSA decreased with the increase in the amount of periodate, thereby improving the thermal stability and dispersion of crosslink collagen fiber [34]. Broderickt et al. synthesized butyl alginate by esterifying sodium alginate and butanol using concentrated sulfuric acid as a catalyst, and successfully prepared stable modified sodium alginate with excellent hydrophilicity by adding butyl to sodium alginate [35]. As a natural polysaccharide, sodium alginate itself has an excellent bacterial affinity. Kai et al. prepared a new chitosan benzalkonium chloride complex with ionic gel, and mixed it into the sodium alginate membrane, improving the antibacterial and mechanical properties of the sodium alginate membrane.

Therefore, based on the phosphoric acid structure of phytic acid from natural plants and the guanidine structure of DL-arginine from animals and plants [36,37,38,39], this paper prepared a biological flame retardant antibacterial agent (PADL) with a high phosphorus content and antibacterial structure. The bio-based composite sodium alginate fiber was prepared by wet spinning after blending and dispersing with alginate, and its structure and properties were characterized. The SA composite fiber is an ideal candidate for application in the textile field because of its flame retardancy and antibacterial properties. In the current research field, Xu et al. prepared a composite SA with an LOI of 30% [40]. Luo et al. prepared SA with an antibacterial rate close to 25% [21]. In comparison, we prepared an SA composite fiber with an LOI of 40% and an antibacterial rate close to 30%, which has a good application prospect in the textile field.

## 2. Materials and Methods

### 2.1. Experimental Raw Materials

SA (Mw = 222,000, DP = 200), (phytic acid and SA, Sinopharm Chemical Reagent Co., Ltd., Beijing, China). (DL-arginine and calcium chloride, Shanghai Titan Technology Co., Ltd., Shanghai, China). Deionized water was used for all experiments and the resistance was 18.2 MΩ.

### 2.2. Preparation of Bio-Based Flame Retardant and Antibacterial Agent

Take 80 g of the phytic acid solution, dilute it to about 30 wt% with deionized water, place it in a three-necked flask, and then take 25.3 g of DL-arginine in the flask according to the molar ratio (PA:DL-arginine = 1:12), and wait for its dissolution.

The flask mentioned above was heated at a constant temperature of 80 °C and continually stirred under the protection of N_2_. The arginine aqueous solution was slowly added to the phytic acid solution at a rate of 2 drops/second, and the feeding was stopped when it was fully reacted until the pH of the solution was close to about 7, and it was left to continue to react for 3 h to terminate the reaction.

Finally, the reaction product was purified by vacuum dehydration and dried in a constant temperature oven at 100 °C, and finally, the obtained sample was sealed and stored.

### 2.3. Preparation of SA Composite Fiber

Prepare 0 wt%, 5 wt%, 10 wt%, 15 wt% and 20 wt% flame retardant PADL aqueous solution, respectively, then take the PADL aqueous solution and 5 wt% SA to mix and stir at a high rate at room temperature until the seaweed sodium is wholly dissolved in the aqueous system to obtain spinning dope.

Take 20 g of the above solution and place it evenly on the tetrafluoro board and carry out surface air drying treatment, and then obtain a series of composite SA film products with different mass fractions and place it in the average temperature environment to wait for its crosslink and curing. Finally, the samples are placed in an oven at 35 °C for surface drying.

Take the prepared spinning dope of different mass fractions using a wet spinning machine to obtain composite SA fibers through a series of processes, such as wet spinning, drying, heat treatment and winding, as shown in Figure 1.

### 2.4. Characterization

#### 2.4.1. Transform Infrared Spectroscopy (FTIR)

In the process of sample preparation, the KBr tableting method was adopted, the samples were mixed with potassium bromide and ground into a powder, and then the samples were compressed and tested. The test was carried out on a Nicolet 8700 FTIR spectrometer(Nicolet Corporation, Madison, WI, USA) with the wavenumber range set to 500–4000 cm^−1^.

#### 2.4.2. Nuclear Magnetic Resonance (NMR)

Took deuterated water (D_2_O) as a solvent, weighed 200 mg of PADL sample, then added deuterated solvent to dissolve. Used Advance 600 nuclear magnetic resonance spectrometer (Bruker, Ettlingen, Germany) to test at room temperature, and its spectral frequency was 600 MHz.

#### 2.4.3. Scanning Electron Microscopy (SEM)

SU−8010 SEM (Hitachi, Kanto, Japan) was used to observe the morphology and dispersion degree of PADL after gold spraying.

#### 2.4.4. Energy-Dispersive X-ray Spectroscopy (EDX)

Quantax 400 EDX (Bruker, Ettlingen, Germany) was used to observe the morphology and dispersion degree of PADL after gold spraying

#### 2.4.5. Thermogravimetric Analysis (TGA)

TGA was carried out on a TG209F1 thermogravimetry analyzer (Netzsth, Freising, Germany). The TGA curve of the sample was collected at a temperature range of 50–800 °C under a nitrogen flow of 30 mL/min. The heating rate was 20 °C/min.

#### 2.4.6. Antibacterial Performance Test

Through the OD value method, Staphylococcus aureus was used to conduct the antibacterial test on the SA composite sample.

#### 2.4.7. Mechanical Property Test

The mechanical properties of composite sodium alginate fiber were tested by a YG006 electronic single-fiber strength tester (Dahe Corporation, Ningbo, China). The sample was tested at an elongation rate of 20 mm/min and pretension of 0.02 cN/dtex. Each sample was measured 10 times and the average value was calculated for the result analysis.

#### 2.4.8. Flame Retardant Test

The SA composite fiber and pure sample were tested for vertical combustion. The UL−1581 horizontal and vertical combustion tester (ASTM Corporation, Beijing, China). The ASTMD635 test standard was adopted, the ignition gas was butane, and the sample specification was 300 mm × 80 mm.

The SA composite fiber and pure samples with different mass fractions were tested for limiting oxygen according to the 5801A limiting oxygen index tester (ASTM Corporation, Beijing, China). The standard adopted was GB/T 2406.2-2009, the ignition gas was butane, and the sample specification was 150 mm × 58 mm.

Used the Vouch−6810 cone calorimeter (FTT, Crawley, UK) to adopt the micro cone calorimetry method, referring to E1354/ISO 5660 standard, and the sample preparation specification was 105 mm × 105 mm × 2 mm, ignite with pulse electric spark, and the sample tested with thermal radiation power of 35 KW/m^2^.

The carbon layer samples obtained were tested by inVia Reflex Raman spectrometer (Renishaw, London, UK) with the scanning range set to 500–3000 cm^−1^ (wavelength: 533 nm, scanning range: 500–3000 cm^−1^).

The combustion cracking mechanism of modified sodium alginate and its pure sample was analyzed with PE TGA400 TG-FTIR combined instrument (PerkinElmer, Waltham, MA, USA). The cracking temperature was 50–800 °C, and the heating rate was 20 °C/min.

## 3. Results

### 3.1. Structural Characterization of Bio-Based Flame Retardant Antibacterial Agents

The distinct groups related to the raw materials and products were detected by infrared and nuclear magnetic resonance, as shown in Figure 2a. The absorption peak at 3360 cm^−1^–3178 cm^−1^ corresponds to the stretching vibration peak of hydroxyl, and the new peak at 1470 cm^−1^ corresponds to the bending vibration peak of −NH^2+^ [41]; the absorption peaks at 1634 cm^−1^, 1404 cm^−1^ and 1167 cm^−1^ are C=O, C−N, and P=O stretching vibration peaks, respectively, and the stretching vibration of P−N−C leads to a new peak at 1072 cm^−1^ [42]. It indicated that the phosphate group of PA reacted with the amino group of arginine to produce a new compound containing the P−N−C group, which successfully proved the synthesis of the target product.

As shown in Figure 2b, the peak between 4.06 ppm and 4.33 ppm is the peak of the phosphate group in the phytic acid molecule. Due to the shielding effect of the amino group in the arginine-deuterated aqueous solution, 1.51 ppm, 3.10 ppm and 3.19 ppm are the peaks of the methylene groups in DL-arginine. The new peak at 3.46 ppm for PADL is mainly due to the C−H in R_2_CH−NH^2+^ formed during the synthesis process [43], indicating that phytic acid and DL-amino acid form a new phytate compound PADL.

XPS was used to analyze the elemental structure of raw materials and products. It can be found that there is also a wave peak of element P compared with DL-arginine in Figure 2c. In addition, Figure 2d,e shows that the peak position shift of N−H and N−C bond binding energy indicates that the binding mode has changed. The peaks at 400.3 eV and 399.6 eV correspond to N−H groups of DL-arginine and PADL, respectively, while the peaks at 401.2 eV and 400.6 eV are N−H and N−C groups before and after the reaction. In addition, the content of N−H in the product PADL is significantly reduced compared with DL-arginine, and new N−P groups (398.6 eV) are generated [44]. It indicates that the introduction of phytic acid can make it react with DL-arginine to form the end product phosphate containing P−N groups.

### 3.2. Morphology and Structure of SA Composite Fiber

After the surface of the composite fiber was sprayed with gold, SEM observed its surface morphology. As shown in Figure 3a, the surface of the pure SA fiber was round, without damage and fracture, and had an excellent straight orientation. When the flame retardant PADL is added, there are gradually regional deposits on the fiber surface. With the increase in flame retardant content, the density of the fiber surface deposits also gradually increases, and the flame retardant is uniformly dispersed on the fiber surface. After magnifying the radial surface by a factor of 1000, it can be observed that the SA composite fiber is entirely covered by flame retardant, indicating that the PADL and alginate are thoroughly mixed in the spinning dope.

SEM-EDX studied the elemental composition of the SA composite fiber samples. The EDX spectra of pure SA and blended SA are shown in Figure 3b, indicating that pure SA is only composed of C, H, O and other elements. In contrast, the blended fabric contains P, N and other elements, which confirms that a large amount of PADL flame retardant has been introduced into the SA matrix. With the increase in PADL, the mass fraction and distribution area of bio-based PADL gradually increased, and the distribution of elements in the matrix was relatively uniform.

### 3.3. Thermal Stability of SA Composite Fiber

Under a nitrogen atmosphere, the thermogravimetric analysis of the SA composite fiber was carried out, shown in Figure 4a,b, and the relevant data are shown in Table 1. The characteristic degradation temperatures were analyzed, including the initial cooling temperature of the sample (T_onset_, mass loss of 5 wt%), the final thermal degradation temperature (T_SECON_), the maximum thermogravimetric temperature (T_MAX_) and the 800 °C residual carbon mass as shown in the TGA and DTG curves. The initial degradation temperature of composite and pure SA was almost the same, and the pyrolysis temperature was around 230 °C, being the main pyrolysis area of SA; a second pyrolysis area appeared between 270 °C and 280 °C, being the central pyrolysis zone of the flame retardant, which promoted char formation of the matrix. The third thermal degradation zone is between 360 °C and 380 °C, this zone is the further decomposition of the substances produced in the second stage, and apparent mass loss occurs during the degradation process. With the phosphorus-based compounds’ pyrolysis and dehydration into carbon, the residual carbon residue rate of the SA composite fiber increased. It avoids heat penetration into the underlying matrix, delaying the diffusion of oxygen and preventing flammable substances from being produced in the first volatilization stage.

It can be seen from the pure SA’s melting curve has a melting peak at about 100 °C and the prepared PADL has a melting peak at about 165 °C in the Figure 4c. The composite sodium alginate fiber with different proportion of PADL has two melting peaks, and two melting peaks are the closest when the addition of PADL is 5%. This shows that SA and PADL have good compatibility when the addition of PADL is 5%. This may be due to the formation of hydrogen bond between −NH_2_ in PADL and −OH in SA. The existence of hydrogen bond promotes the compatibility of PADL with the matrix. However, with the further increase of PADL addition, it will appear obvious agglomeration in the spinning solution. PADL cannot be uniformly dispersed in the fiber, resulting in poor compatibility, which is shown by the large difference between the two melting peaks in the DSC curve.

### 3.4. Antibacterial Properties of SA Composite Fiber

In order to analyze the bactericidal ability of the SA composite fiber samples, all the samples to be tested were co-cultured with Staphylococcus aureus for 6 h, and the antibacterial ability of all samples was tested using the OD value method to evaluate the antibacterial activity of the SA composite fiber. The microbial OD value is an indicator reflecting the growth state of bacteria. OD is the abbreviation of optical density, which represents the optical density absorbed by the tested substance. Generally, the range of microbial measurement is 400–700 nm, and the maximum absorption wavelength needs to be measured by an ultraviolet spectrophotometer [45]. As shown in Figure 5, pure SA has a specific antibacterial effect, and with the increase in the mass fraction of PADL, its antibacterial performance also increases. When the addition amount reaches 20 wt%, the antibacterial rate can be increased by 30%.

PA can inhibit the growth of Staphylococcus aureus by inducing excessive permeability in the cell membrane, which causes changes in cell morphologies and decreases intracellular ATP concentration. The arginine structure contains a guanidine group component. The −NH^3+^ formed in the solution is a charged functional group, which has a significant interaction with the cell membrane. It can penetrate the cell to adsorb intracellular anionic groups, interfere with the cell’s metabolic activity and kill bacteria [46].

### 3.5. Mechanical Properties of SA Composite Fiber

Stress–strain curves were obtained after tensile testing of SA fiber with different PADL loadings. As shown in Figure 6 and Table 2, the pure SA fiber has a high degree of strain, and the elongation at break can reach 32.9%. However, the SA fiber’s tensile strength is relatively low, only reaching 6.8 MPa, and the elastic modulus is also low, only 0.5 MPa. However, the tensile strength and elastic modulus of the fiber increased significantly after loading flame retardant PADL. The SA composite fiber’s tensile strength and elastic modulus are the highest when the load is 5 wt%. −OH in sodium alginate can form a conjugated hydrogen bond with −NH_2_ in PADL, which increases the strength of the fiber. When the loading amount is 5 wt%, the tensile strength of the composite fiber is the highest, close to 12.2 MPa, indicating good compatibility between the flame retardant and the composite fiber. However, the tensile strength and elastic modulus of SA composite fiber gradually decreased as the load of flame retardant PADL continued to increase. When the amount of PADL is increased, it will be agglomerated in the spinning solution, and cannot be uniformly dispersed in the fiber. Therefore, the mechanical properties of the composite fiber are weakened, resulting in the gradual decrease in tensile strength and elastic modulus.

### 3.6. Combustion Behavior and Flame Retardant Mechanism of SA Composite Fiber

The LOI of the material is an essential indicator of flammability. As for the LOI, the material cannot burn until oxygen concentration reaches a limiting value.

Figure 7 and Table 3 give the values of the LOI and vertical combustion test for SA composite fiber. An observation can be made that with the increase in the additional amount, the burning time gradually shortened, and the LOI gradually increased. At the same time, the char formation also gradually increased, the length of the carbon layer decreased with the increase in the additional amount and the flame retardant performance significantly improved. When the addition amount reaches 20 wt%, the flame retardant effect is the best, the LOI is 40 and the product is V−0 grade.

As a functional bench-scale test, the cone calorimeter test (CCT) is a typical technique for simulating actual fire situations on a lab scale, which is a significant factor in the quantitative analysis of flammability. To further investigate the flame-resistant property of SA composite, all samples were characterized by CCT. Heat release rate (HRR), peak heat release rate (PHRR), time to ignite (TTI), total heat release (THR), and mass loss (%) were obtained from CCT.

Figure 8 and Table 4 give the value of the cone calorimeter test for SA composite fiber. The heat release rate of 20%PADL is the lowest, and the whole process tends to be zero. The heat release rate of 15%PADL is second only to it, and tends to zero around 75 s. The total heat release of 15%PADL is much lower than that of smaller PADLs. However, smoke production rates of 20%PADL and 15%PADL tended to zero at around 75 s, which was higher than that of other PADLs with a smaller proportion, and their total smoke production was much higher than that of other PADLs with a smaller proportion. This is because the pyrolysis of PADL forms a barrier layer that inhibits the production of flammable substances from the matrix when the temperature increases. With the increase in the amount of PADL added, the smoke density increased. When the amount of PADL reached 20 wt%, the TSR value increased from 2.08 to 184.3 (m^2^/m^2^). This phenomenon is related to the PADL content deposited on the surface of the SA film. The flame retardant PADL will form protective carbon when the surface burns, but at the same time of burning, with the increase in the added amount, due to its combustion of CO, the content of smoke such as CO_2_ also increases gradually.

The carbon residue morphology was analyzed on the SA composite fiber after combustion, and the results are shown in Figure 9. Pure SA shrinks after being burned at high temperatures, while SA composite fiber keeps its original state. With the increase in PADL, the residual carbon is denser, and a particular expansion effect appears during combustion. Some tiny bubbles can be observed from the surface of the carbon residue, which is caused by gas escape during the combustion process, further illustrating its expansion effect. The use of its expanded carbon layer can effectively isolate the external combustion-supporting substances and act as a cohesive phase resistance.

Infrared analysis of the burned SA composite fiber results is shown in Figure 10a. Values of 3308 cm^−1^, 1641 cm^−1^ and 1381 cm^−1^ correspond to the stretching vibration absorption peaks of hydroxyl, C=O and C=N, respectively, while the stretching vibration peaks of P=O in the flame retardant PADL are located at 1114 cm^−1^. In addition, a gradually enhanced absorption peak appears at 1037 cm^−1^: the stretching vibration peak of the P−N−C structure in the compound. In summary, PADL pyrolyzes out phosphate groups under high-temperature conditions, so SA’s carbon-forming ability is gradually enhanced [47].

The structure of the SA composite fiber carbon residue was tested by Raman, as shown in Figure 10b–f. The peak fitting mode uses a Gaussian model. The G peak appears at 1580 cm^−1^, representing the organized graphitization structure of the carbon layer, the D peak at 1350 cm^−1^ corresponds to the degree of disorder of the carbon structure in the material and the intensity ratio of D to G is related to the microscopic degree of the carbon layer. It is related to the crystal size. The larger the value, the smaller the carbonaceous structure and the higher the protective shielding efficiency. The increase in the ID/IG value shows that the higher the order of the carbon layer, the higher the density of the carbon layer and the better the flame retardant performance. The spectra show that the ID/IG ratio increases from 0.96 to 1.33, and the degree of graphitization of the carbon layer increases with the increase in PADL addition. PADL promotes SA to form expanded carbon with increasing density at high temperatures and volatilizes a large amount of non-combustible gas. The expansion carbon formed by the synergistic effect of P and N and gas dilution improves the flame retardancy of SA composite fiber.

The flame retardant mechanism of SA composite fiber was explored by thermogravimetric–infrared spectroscopy (TG-IR). Figure 11 shows the TG and FTIR spectra of composite SA in a nitrogen atmosphere. There is a significant difference in the peak range of around 1000 cm^−1^ due to the formation of a large number of nitrogen-containing compounds with C−N and C=N bonds during the pyrolysis process of SA composite fiber. When the temperature of pure SA rises to 800 °C, a vibration peak with greater intensity appears in the characteristic band around 2200 cm^−1^, which is the carboxy group formed by the further decomposition of carbon-containing compounds. The peaks at 3550–3700 cm^−1^, 2300 cm^−1^ and 1300–1800 cm^−1^ can be attributed to gaseous products such as the carboxyl group, aldehyde group and hydrocarbon, respectively. This also shows that PADL causes the SA composite fiber to decompose and produce non-flammable nitrogen-containing compounds to achieve the flame retardant effect.

## 4. Conclusions

Flame retardant and antibacterial SA fiber was fabricated using the bio-based flame retardant of phytic acid and DL-arginine successively. Then the morphological structures, combustion behavior, thermal stability and mechanical as well as antibacterial properties of the fiber were investigated carefully. The prepared bio-based flame retardant PADL can effectively improve the flame retardant properties of the SA fiber and can be well dispersed in the SA matrix. In the vertical combustion and limiting oxygen experiments, the flame retardant properties of the SA composite fiber were significantly improved with the increase in the amount of PADL added. When the additional amount of PADL in SA composite fiber is 20 wt%, its LOI is 40.0 ± 0.3% and UL−94 is V−0 grade. The carbon residue after combustion was tested by SEM, infrared, Raman, etc. It was found that PADL can promote surface carbonization, and has a specific expansion effect, which improves the carbon layer compactness and graphitization degree of SA. At the same time, the SA composite fiber also has specific antibacterial and mechanical properties. When the additional amount of PADL in SA composite fiber is 20 wt%, the antibacterial property of the fiber is increased by 30%.

## Figures and Tables

**Figure 1 polymers-15-00154-f001:**
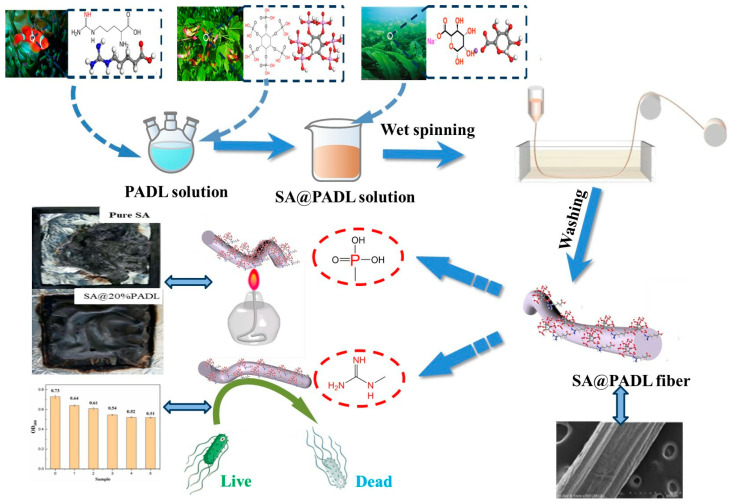
Schematic illustration of the preparation of SA composite fiber by wet spinning.

**Figure 2 polymers-15-00154-f002:**
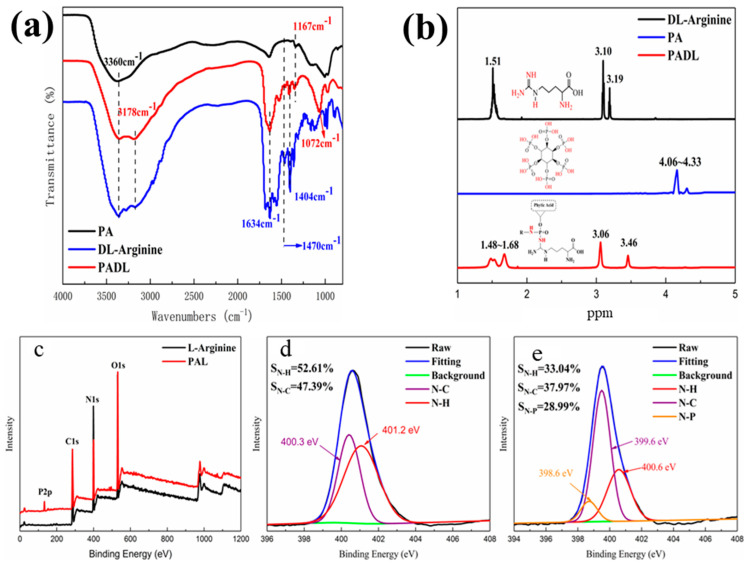
(**a**) Infrared spectra of PADL, phytic acid and DL-arginine; (**b**) ^1^H NMR spectra of DL-arginine, phytic acid and PADL; (**c**–**e**) the XPS spectra of PADL, PA and DL-arginine.

**Figure 3 polymers-15-00154-f003:**
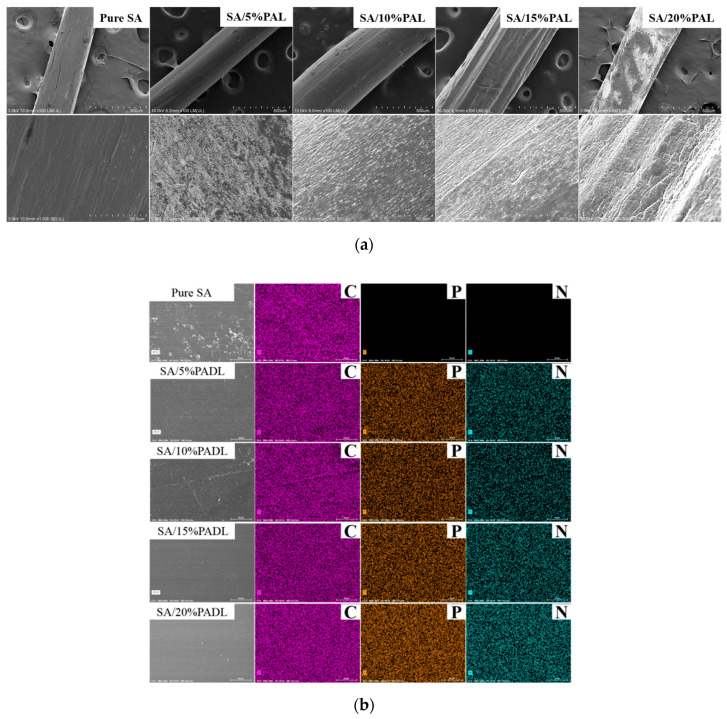
(**a**) The SEM image of the side SA composite fiber; (**b**) EDX scan spectrum of the SA composite fiber side.

**Figure 4 polymers-15-00154-f004:**
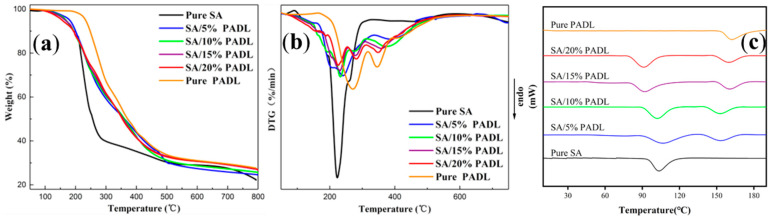
(**a**) TGA, (**b**) DTG and (**c**) DSC curves of PADL and SA composite fiber.

**Figure 5 polymers-15-00154-f005:**
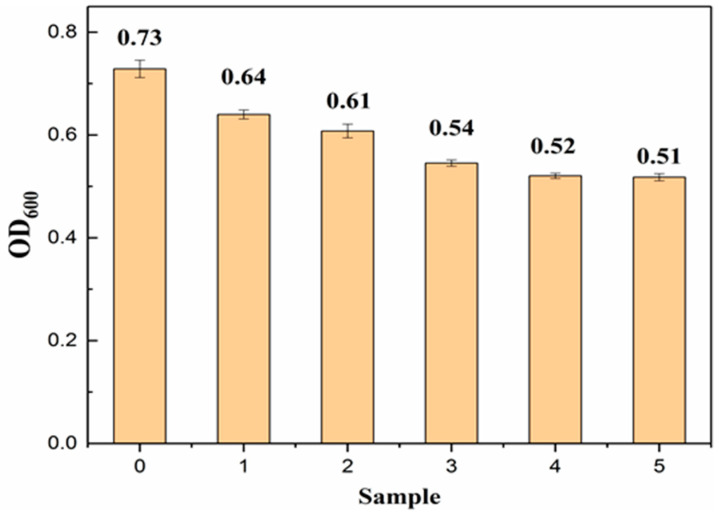
Antibacterial spectrum of SA composite fiber (0: blank control; 1: pure SA; 2: SA/5 wt%PADL; 3: SA/10 wt%PADL; 4: SA/15 wt%PADL; 5: SA/20 wt%PADL). The mechanical properties of SA composite fiber.

**Figure 6 polymers-15-00154-f006:**
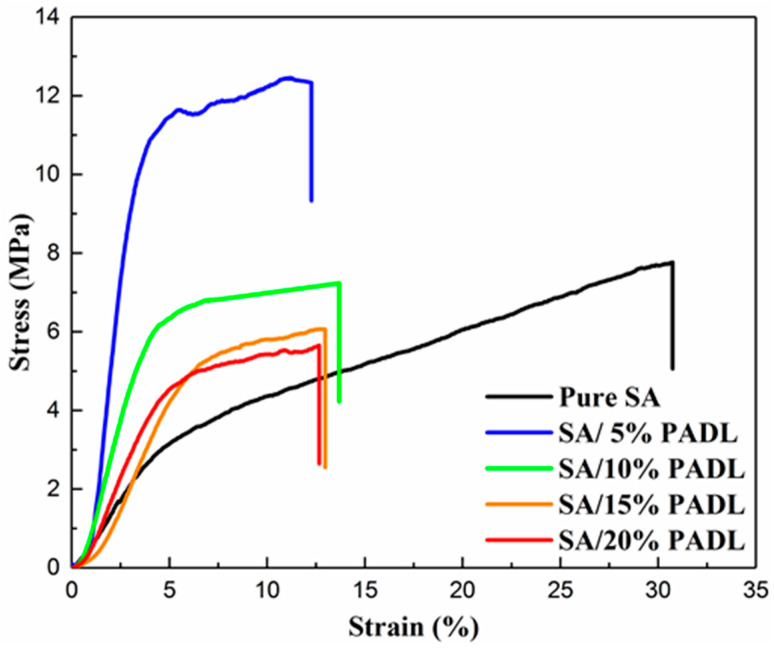
The mechanical properties of SA composite fiber.

**Figure 7 polymers-15-00154-f007:**
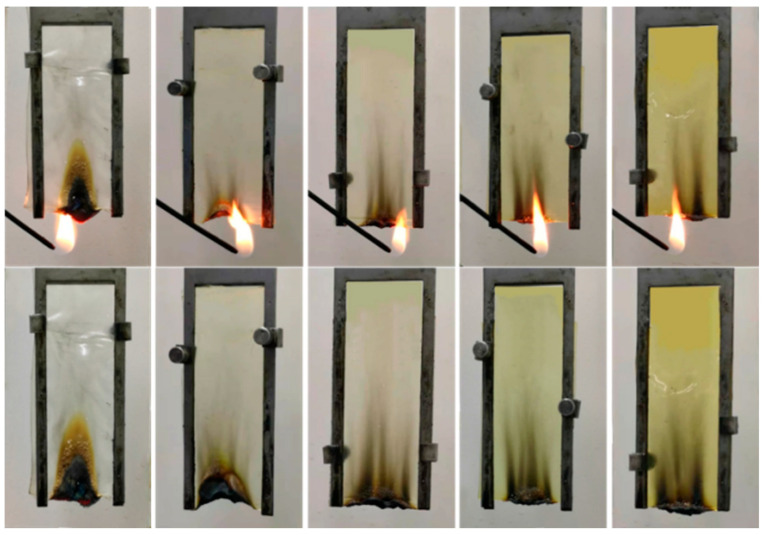
Real-time picture of vertical combustion test.

**Figure 8 polymers-15-00154-f008:**
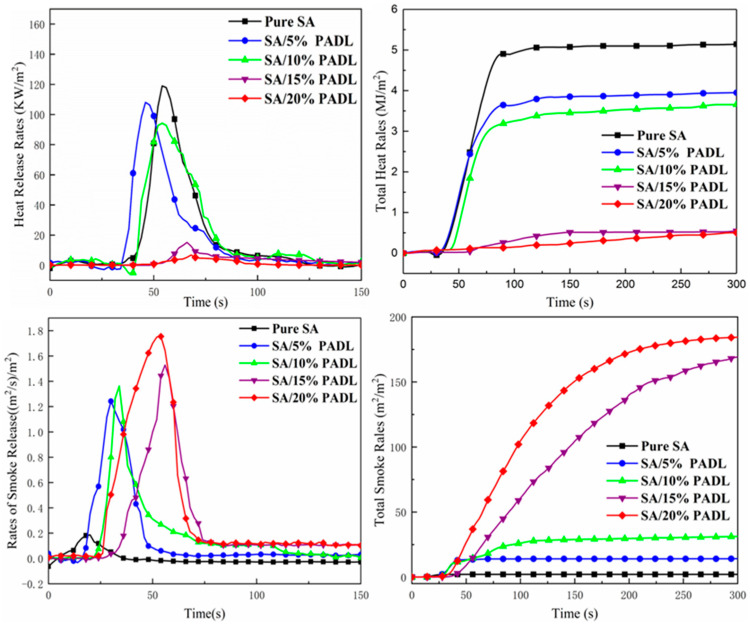
The HRR/THR and RSR/TSR curves of SA composite fiber.

**Figure 9 polymers-15-00154-f009:**
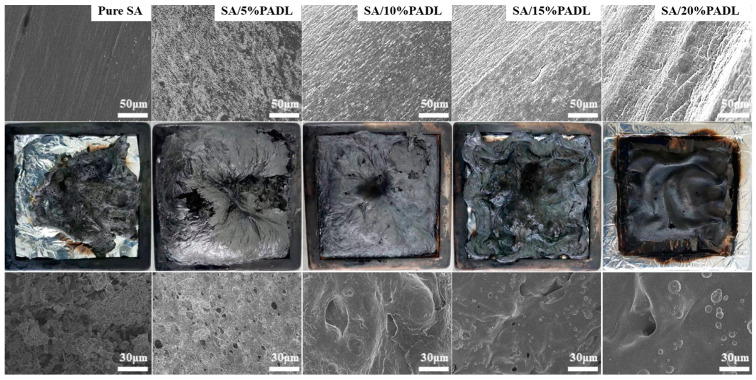
SEM micrographs of char residue surface of SA composite fiber.

**Figure 10 polymers-15-00154-f010:**
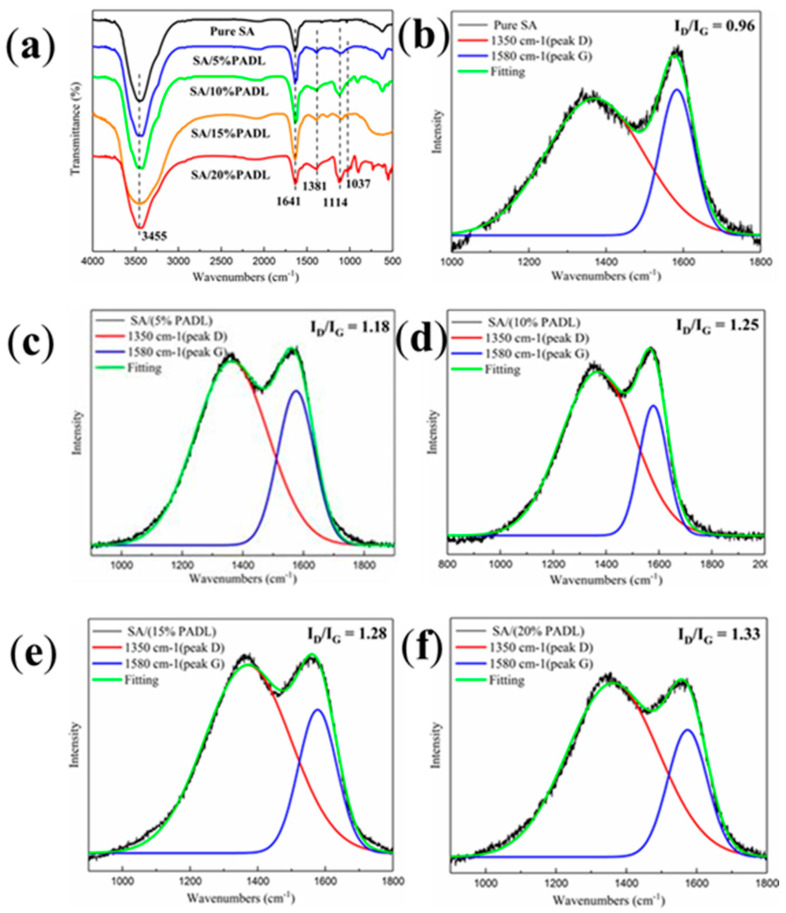
(**a**) The FTIR spectra of char residue of SA composite fiber; (**b**–**f**) the Raman spectra of char residue of SA composite fiber.

**Figure 11 polymers-15-00154-f011:**
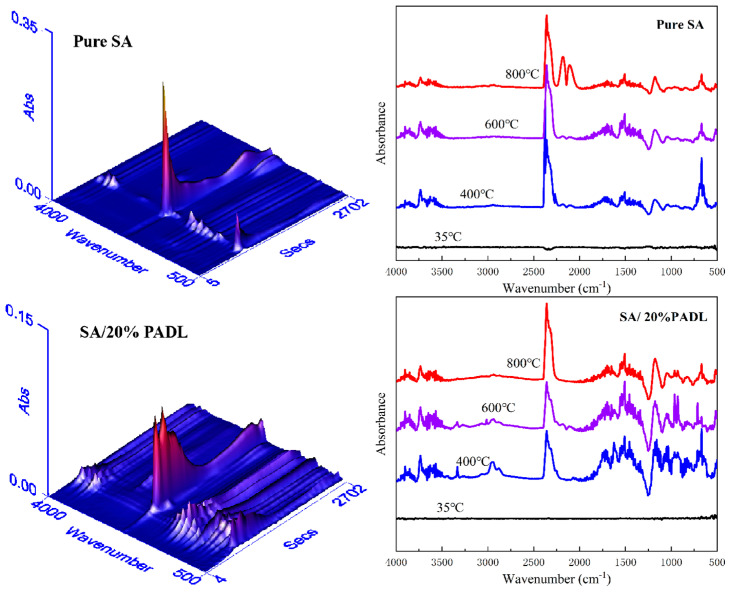
Three-dimensional TG-IR spectra and FTIR spectra of gasified pyrolysis products for SA and SA/20%PADL.

**Table 1 polymers-15-00154-t001:** TGA-DTG data of PADL and SA composite fiber.

Sample	T_ONSET_ (°C)	T_SECON_ (°C)	T_MAX_ (°C)	DTG_max_	Residual at 800 °C (wt%)
Pure SA	223	/	/	−1.03	20.6
SA/5% PADL	239	275	381	−0.39	24.4
SA/10% PADL	232	276	370	−0.39	25.3
SA/15% PADL	230	277	364	−0.36	26.6
SA/20% PADL	228	283	348	−0.33	26.7
Pure PADL	271	347	/	−0.48	28.9

**Table 2 polymers-15-00154-t002:** The mechanical properties of SA composite fiber.

Sample	Elastic Modulus (MPa)	Tensile Strength (MPa)	Elongation at Break (%)
Pure SA	0.5	6.8	32.9
SA/5%PADL	4.2	12.2	12.2
SA/10%PADL	1.9	7.2	13.7
SA/15%PADL	1.1	6.1	12.9
SA/20%PADL	1.1	5.6	12.6

**Table 3 polymers-15-00154-t003:** Results of LOI and vertical combustion test.

Sample	LOI (%)	Vertical Combustion Test		
		After Flame Time (s)	After Glow Time (s)	Char Length (cm)
Pure SA	26.0 ± 0.2	4.0	3.2	6.0 ± 0.3
SA/5 wt%PADL	30.0 ± 0.1	3.2	0	4.8 ± 0.1
SA/10 wt%PADL	34.0 ± 0.1	3.0	0	3.2 ± 0.1
SA/15 wt%PADL	37.0 ± 0.3	1.7	0	1.3 ± 0.2
SA/20 wt%PADL	40.0 ± 0.2	0	0	0.8 ± 0.1

**Table 4 polymers-15-00154-t004:** The HRR/THR and RSR/TSR data of SA composite fiber.

Sample	TTI (s)	Peak HRR (KW/m^2^)	THR (MJ/m^2^)	Peak RSR ((m^2^/s)/m^2^)	TSR (m^2^/m^2^)	Char Residues (%)
Pure SA	21	119.0 ± 0.3	5.14 ± 0.05	0.18 ± 0.05	2.08 ± 0.05	23.0 ± 0.1
SA/5% PADL	32	108.0 ± 0.1	3.95 ± 0.01	1.24 ± 0.01	14.13 ± 0.01	25.7 ± 0.5
SA/10% PADL	34	94.2 ± 0.2	3.65± 0.02	1.36 ± 0.06	31.23 ± 0.04	34.6 ± 0.5
SA/15% PADL	48	15.2 ± 0.1	0.54 ± 0.04	1.52 ± 0.07	169.2 ± 0.01	36.0 ± 0.1
SA/20% PADL	52	6.7 ± 0.2	0.50 ± 0.01	1.75 ± 0.02	184.3 ± 0.03	44.2 ± 0.2

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
