# Peer review of "Preparation and Mechanism of Bio-Based Sodium Alginate Fibers with Flame Retardant and Antibacterial Properties"

_polymers, 2022, doi:10.3390/polym15010154_

Round 1

Reviewer 1 Report

This manuscript presents valuable experiment with all obtained data, but it is poorly presented, omitting scientific style and some parts are even confusing. 

 Title is not clear. Part "with flame retardant and antibacterial" is not well defined. Antibacterial what?

 Please provide exact values of used reagents through the Part 2 (Materials and methods). "Take a certain molar fraction/Take a certain amount/..." is inadmissible to remain in entire Part 2. Please avoid using imperative style. All equipment should be named like "plasma emission spectrometer (Prodigy, Leeman, USA) - name the equipment (producer, country of origin).

 Line 148: OD value method. Explain method or refer to adequate reference

 Line 237-246: Give the meaning of LOI abbreviation

 Part 3.4.: Why did you put these results together?!

 When presenting the results, compare them with available literature. This is "the weakest" point of the manuscript. Only Part 3.5. is well explained and written with underlaying mechanisms. 

 In conclusion, the manuscript has a quality methodology and valuable results, but the discussion of the results must be improved by comparing results to the existing literature, as the entire section Materials and methods must be written more thoroughly.

Reviewer 2 Report

Dear Authors,

I studied your manuscript entitled "Preparation and mechanism of bio-based sodium alginate fibers with flame retardant and antibacterial". This paper comprises interesting results that certainly deserve publication. I recommend a minor revision before further consideration for publication in the Polymers.

1) The research question needs to be more well-stated and discussed. The recent literature review should be summarized for benchmarking purposes and discussed in detail with your research findings.

2) This manuscript has a phenomenological style, observing a result and explaining it with statements. It would be helpful if you conducted more analysis based on published research.

3) Figures 3 and 8: The scale bars on the SEM images are not visible. Add a decent-sized scale bar for all images.

4) Please provide manufacturer details (model, city, or country) for all characterization instruments.

5) It would also be beneficial if the manuscript included a brief comparison between wet-spun fibers and electrospun fibers. Some related papers were suggested to be used:

a) https://doi.org/10.1016/j.progpolymsci.2020.101346

b) https://doi.org/10.3389/fbioe.2022.1027351

c) https://doi.org/10.3390/polym14163266

6) English language needs some polishing since some terms are vague. The paper's title is also recommended to be revised.

Reviewer 3 Report

Dear authors,

My comments on this paper are the following:

1. Which is the novelty of the paper with respect to existent literature?

2. XPS analysis could be added.

3. The title should be adjusted: "...with flame retardant and antibacterial" properties or behaviour.

4. Abstract: pay attention to English: "Flame retardant and antibacterial sodium alginate(SA) fiber were fabricated..." fibers or was fabricated.

5. 2.2. Instead of speed you could use rate.

6. Which are the characteristics of sodium alginate used in this study (MW, DP etc.)?

7. Describe the crosslinking mechanism.

8. Discuss the advantages of these fibers for the target application.

9. DSC analysis should be added to check the influence of the additives (flame retardant for example) on the characteristic temperatures of sodium alginate.

10. Discuss the tensile test results taking into account the application and similar papers in literature. Show the values for the elastic moduli.

11. English check up should be done throughout the paper.

Round 2

Reviewer 1 Report

All the objections were accepted and the manuscript has been improved. Only one thing is missing - the authors gave explanation in Response toReviewer regarding objection 1 (related to the title change and the addition of word "properties"), but the change hasn't been done in the Manuscript itself (in the revised version).

Reviewer 3 Report

Dear authors,

Your paper has been improved. The title seems now more appropriate.

Minor changes are required:

1. English should be checked again. For example: Introduction - "Physical modification and chemical modification are commonly used modification methods of sodium alginate at present".  Physical and chemical modification are...

2. "Compared with the existing products in the textile field, the SA composite fiber prepared in this research not only has excellent flame retardant and antibacterial properties, but also is a full biological material, which meets the requirements of green and environmental protection". A reference for these requirements could be added here.

3. 2.1. Materials: SA, MW222. What do you mean by that 222?

4. I would be happy to check the final version of your paper after minor revision.
